# A New Beach Topography-Based Method for Shoreline Identification

**Marco Luppichini** [1,*], **Monica Bini** [2], **Marco Paterni** [3], **Andrea Berton** [3] and **Silvia Merlino** [4]

[1] Department of Earth Sciences, University of Study of Florence, Via La Pira 4, 50121 Florence, Italy

[2] Department of Earth Sciences, University of Pisa, Via S. Maria, 52, 56126 Pisa, Italy; monica.bini@unipi.it

[3] Istituto di Fisiologia Clinica del Consiglio Nazionale delle Ricerche, IFC—CNR, 56124 Pisa (PI), Italy; paternim@ifc.cnr.it (M.P.); andrea.berton@ifc.cnr.it (A.B.)

[4] Istituto di Scienze Marine del Consiglio Nazionale delle Ricerche, ISMAR—CNR, 19032 Lerici (SP), Italy; silvia.merlino@sp.ismar.cnr.it

\* Correspondence: marco.luppichini@unifi.it

**Abstract:** The definition of shoreline is not the same for all contexts, and it is often a subjective matter. Various methods exist that are based on the use of different instruments that can determine and highlight a shoreline. In recent years, numerous studies have employed photogrammetric methods, based on different colours, to map the boundary between water and land. These works use images acquired by satellites, drones, or cameras, and differ mainly in terms of resolution. Such methods can identify a shoreline by means of automatic, semi-automatic, or manual procedures. The aim of this work is to find and promote a new and valid beach topography-based algorithm, able to identify the shoreline. We apply the Structure from Motion (SfM) techniques to reconstruct a high-resolution Digital Elevation Model by means of a drone for image acquisition. The algorithm is based on the variation of the topographic beach profile caused by the transition from water to sand. The SfM technique is not efficient when applied to reflecting surfaces like sea water resulting in a very irregular and unnatural profile over the sea. Taking advantage of this fact, the algorithm searches for the point in the space where a beach profile changes from irregular to regular, causing a transition from water to land. The algorithm is promoted by the release of a QGIS v3.x plugin, which allows the easy application and extraction of other shorelines.

**Keywords:** shoreline; beach; UAV; GPS; Structure from Motion; QGIS plugin; topography

## 1. Introduction

The monitoring of coastal areas is very important and meaningful to safeguard the benefits that these areas bring to the environment and to the human activities. Monitoring starts with the control of the shoreline, which is the line where the land meets the sea. Shorelines are idealized as the dynamic interface between water and land [1] and, according to Boak and Turner [2], there are two main categories of shoreline indicators: those based on the detection or identification of visible features (e.g., instantaneous water lines, vegetation lines) and those based on the intersection of the coastal profile with a specific elevation datum like the 0m Above Mean Sea Level (AMSL). Different methodologies exist for coastal monitoring, which are based on direct and remote acquisition systems. Direct shoreline surveys are normally conducted using the DGPS technique of post-processing or of real-time methodology [3,4]. The main drawback of this method lies in the amount of time required to cover large stretches of coastline. Remote sensing for the correct positioning of the shoreline can be distinguished by observation of satellite images [5–9], Unmanned Aerial Vehicles (UAV) [10–13], video monitoring [14,15], historic aerial photos, and cartography [16,17].

Several techniques designed to discriminate between sea and land have been proposed to extract shorelines from images. Plant and Holman [18] used a method initially developed for grey-scale cameras, called Shoreline Intensity Maximum (SLIM). Recently, with the adoption of colour cameras, spectral information has also been exploited to identify the shoreline, using the water property to absorb the Red signal (R) and the sand property to absorb the Green (G) and blue signals [19].

This work aims at finding a new method based on the beach profile derived from high resolution Digital Elevation Model (DEM) to identify the shoreline. More specifically, we propose a method based on Structure from Motion (SfM) techniques to build a high-resolution DEM from UAV images to identify the position of shorelines. We believe this innovative method can be better applied in various contexts with respect to techniques based on image colours. Such methods are influenced by various factors such as time of acquisition, period of year, cloudy weather conditions, and waves. These factors often strongly influenced the results, while a method based on the topography of the beach was more objective and stable for the specific beach that we studied. Some studies use DEMs, for example LiDARs (e.g., [20,21]), which have a lower resolution than DEMs derived from UAV images for the extraction of the shorelines. However, the limit of these methodologies, which use the profile of the beach to identify the position of the shoreline, rests on the resolution of the DEM; furthermore, the costs due to the types of data employed prevent frequent monitoring of the coast. Other studies use UAV images to extract high resolution DEMs for the study of volume variation over time (e.g [22,23]). With these methodologies, it is necessary to perform further sampling, but the result always yields volume variation. Moreover, the results of these topographical surveys are difficult to compare with those of past measurements carried out using instruments like cartography, DGPS, etc. There is also a difference in volume between sequential DEMs over time; therefore, these methods make it impossible to compare the results with those obtained with other techniques such as shorelines derived from satellites and/or aerial images. The method we propose in this study allows to identify the shorelines more precisely with respect to the use of DEMs, by using LiDARs and to trace a boundary between land and water, unlike the main investigation methods that use DEMs obtained from UAV images.

## 2. Materials and Methods

### 2.1. Study Area

To investigate the role of different methods in the reconstruction of shorelines, we chose to acquire UAV images and Differential GPS (DGPS) in the northern sector of the Pisa coastal plain. The study area is located in central Italy, and in particular in the Tuscany Region. This coastal stretch was chosen because historically affected by erosion and hardly touched by anthropic settlement since it is part of the Migliarino, San Rossore, and Massaciuccoli Regional park. The stretch of coast studied is about 4.5 km and is located on the right bank of the Arno River. The beach studied is sandy (Figure 1). The coast experience microtidal regime with the spring tide is just ca. 30 cm [24].

### 2.2. DGPS Survay

We have sampled 224 points with a R8s Trimble real-time kinematic (RTK) DGPS. The 224 points were divided into pairs, so that the shoreline was sampled by taking a point in the water and a point on the land where the waves ended [25]. Sampling took place in calm waters. We built the shoreline starting from DGPS points and we designed an appropriate algorithm able to automatically place the shoreline between the two points (one towards land and the other towards sea). The operations of this algorithm are illustrated in Figure 2. Positioning of the shoreline depends on the elevation of the two investigated DGPS points. In case one point has a negative elevation and the other point has a positive one, the algorithm makes a linear interpolation between the two acquired points. It selects the point with 0 m of elevation (coordinate z) among the interpolate points and it derives the x and y (latitude and longitude) coordinates of the shoreline from this point. In case both points have positive (or negative) elevations, the only difference is that the algorithm selects along the topography profile the point

(latitude and longitude) with the coordinate z equal to the mean quote of two DGPS points investigated from this point (Figure 2). With this algorithm, we assume AMSL in function of the elevation of the DGPS points. When the 0 m AMSL is included in the topography profile, we consider this quote as the most representative of the margin between water and land. When the 0m AMSL is not included in the topography profile, we must choose another AMSL quote. In such case, we believe that the best representative quote is the mean elevation between the two extremes of the profile (one towards land and the other towards sea).

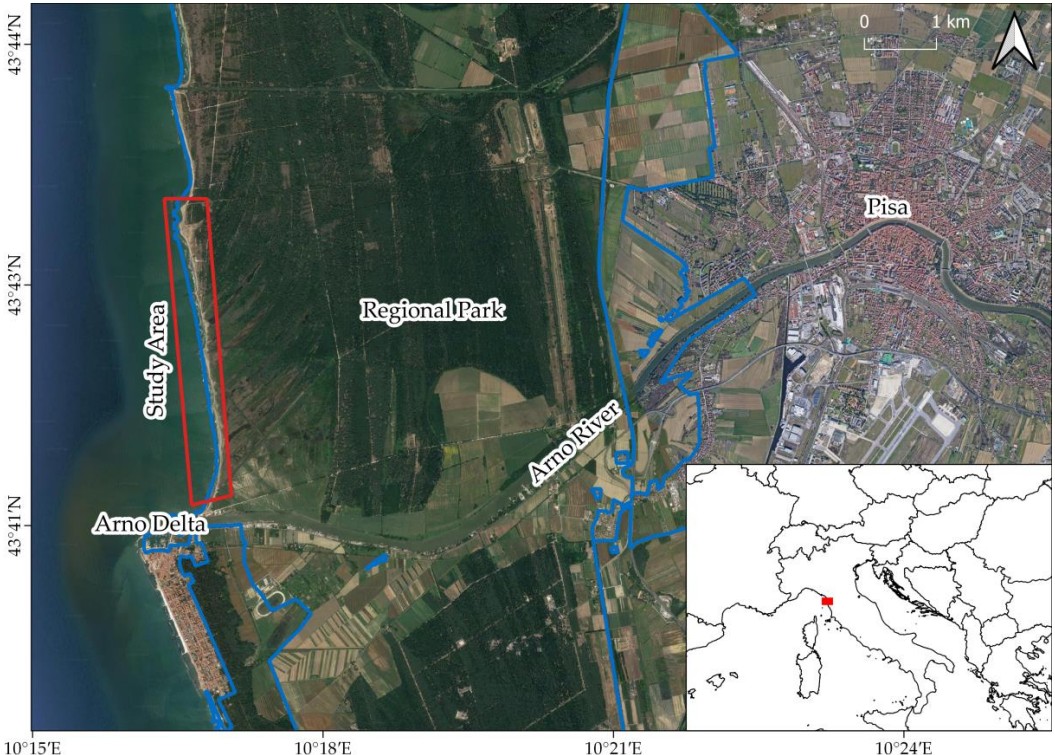

**Figure 1.** Study area.

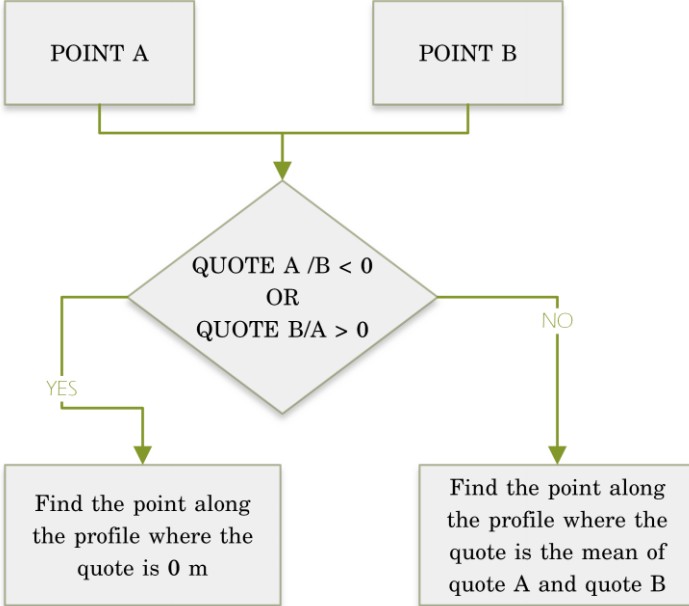

**Figure 2.** Flowchart of the algorithm designed for identification of the shoreline using differential GPS (DGPS) points.

After the identification of all the 112 points of shoreline, we draw the polyline representing the margin between water and land.

### 2.3. UAV Survey

SfM photogrammetry is a technique that allows to reconstruct 3D models starting from a collection of photos of the same elements obtained from different viewpoints [26–28]. The frames are sampled by means of an Unmanned Aerial Vehicle (UAV) equipped with a consumer-grade camera.

In particular, we used DJI Phantom 4 Pro V2, which is a quadcopter with a flight autonomy of 30 min even if, for safety reasons, we did not exceed 20 min of flight. The FC6310S camera was able to take photos of 5472 × 3648 pixels (in a 3:2 aspect ratio setting).

All acquisitions were obtained with a 24 mm focal length and camera oriented in orthogonal mode with respect to the ground.

The flights were in automatic mode and reached a maximum distance of 500 m from the pilot (as required by the Italian regulatory system), making it possible to perform 1 km sections for each flight.

All flight plans were created using the Desktop UgCS (Universal Ground Control Station) software and were performed using the UgCS application for Android OS. The "Area scan" function allowed us to set the parameters so as to obtain a flight height of 50 m above ground level (AGL) and an overlap of the acquired photos equal to 75% for each side. By using this flight height, we were able to scan a 75 × 50-m area for each photo. Image acquisition was directly controlled by the flight execution software, UgCS for DJI (Android version); the shooting interval was set to 2 s, the manual focus to infinity, while disabling the autoexposure, and storage format was JPG. Four parallel transects were performed for each flight, to obtain a mapping of 1000 × 70 m with the yaw of the drone constantly set at the same angle with respect to the Earth's North.

The UAV and DPGS samplings were carried out simultaneously; moreover, the area is subject to a small tidal excursion [24]. For this reason, we can consider that the two samples represent the same situation of the sea.

### 2.4. Data Processing

Georeferencing of the 3D model obtained through SfM requires the identification of the ground control points (GCPs) of already known coordinates. We processed the photos and GCPs by using Metashape Professional software (Agisoft LLC, St. Petersburg, Russia), which implements SfM and multi-view stereo matching algorithms.

The first step in the standard workflow by Metashape Professional is to upload a set of images and to evaluate their quality. Metashape Professional finds correspondence points between overlapping images. It estimates the camera position for each photo and creates a scattered point cloud model. When available, the GCPs are identified in the images, and their coordinates are entered. GCPs are typically used as control points to optimize camera position and orientation data, making it possible to obtain better model reference results. The next step is to create a dense point cloud based on the estimated positions and parameters for each camera. Finally, a DEM and an orthomosaic are calculated with pixel dimensions that depend on the average resolution of original image terrain sampling.

### 2.5. Shoreline Identification Algorithm

To extract the shoreline from DEMs and orthomosaics, we identified a new semi-automatic method based on the beach profile by the SfM technique. The method is based on the principle that SfM performs poorly on uniform or reflecting surfaces like the sea [29]. The beach profiles obtained with SfM are more irregular and unrealistic on sea, becoming regular and realistic when the points are referred to the land. By exploiting this principle, we used an algorithm that sought the transition point between an irregular profile (sea surface) and a more regular profile (beach). The algorithm is based on the use of transects along the beach.

The profile of the transect must start from the surface of the sea and end on land. The profile of the transect, which includes the surface of the sea, will be characterized by a low coefficient of determination ($R^2$; Figure 3), moving from the sea towards the beach and gradually discarding part of the transect's profile, which will be regularized until it includes only and exclusively the beach profile.

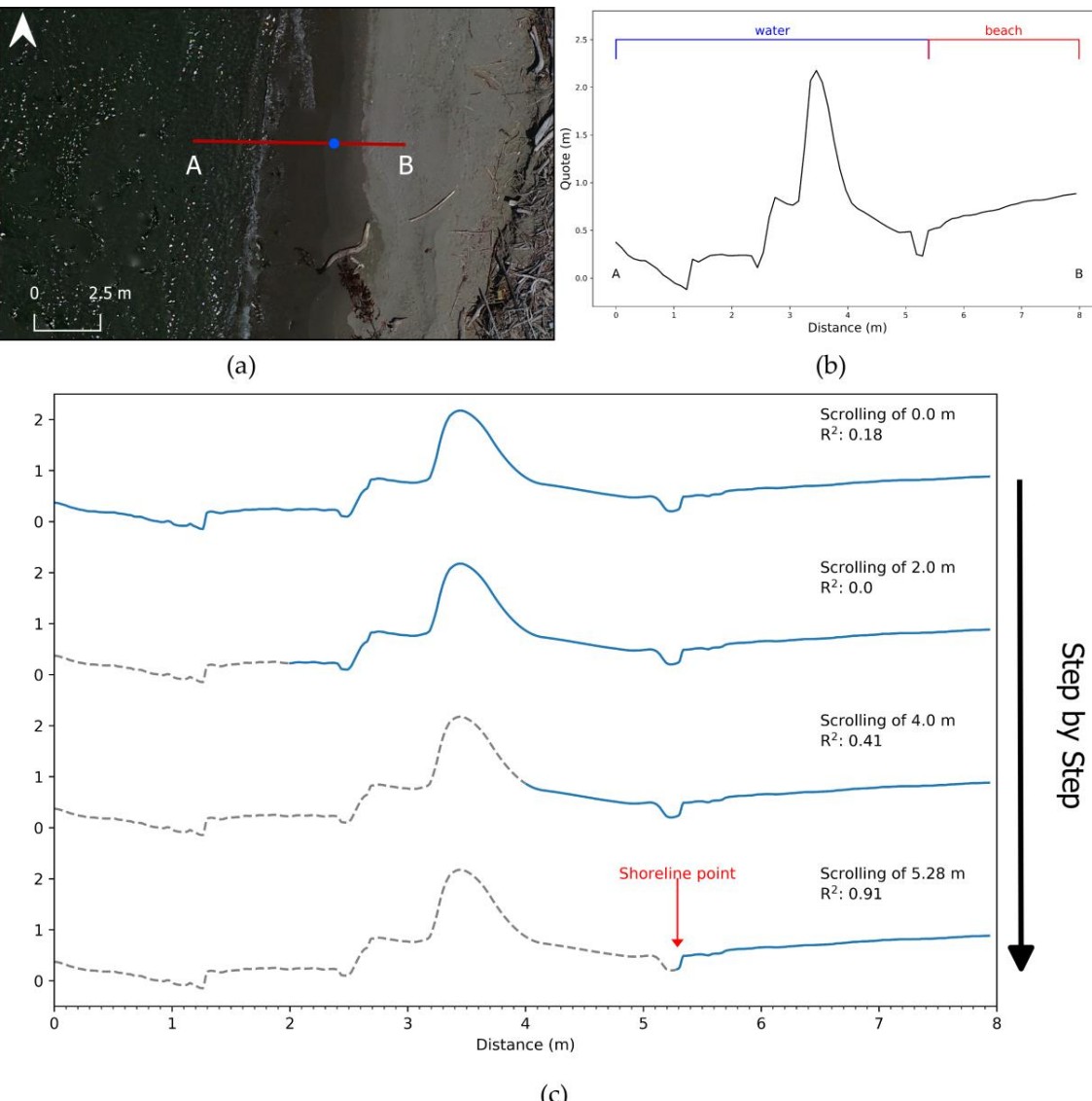

**Figure 3.** Description of shoreline identification from the Unmanned Aerial Vehicle (UAV) orthomosaic and Digital Elevation Model (DEM). (**a**) The location is a typical transect starting from sea and ending on the beach (red line). The blue point represents the local attribution of the shoreline by an algorithm; (**b**) profile on DEM from SfM of the investigated transect; (**c**) four illustrative steps of the algorithm to find the shoreline point. $R^2$ is calculated only on the part of the profile coloured in blue. The grey dashed line represents the progressive part of the profile discarded by the algorithm.

When the profile has an $R^2$ greater than or equal to a determined threshold, the algorithm stops and associates the point of coordinates closest to the sea (for example the shoreline point) (Figure 3). 
We developed a QGIS v3.x plugin in Python 3 to make the algorithm available to the scientific community working in the field. The plugin can be downloaded at the following GitHub link: https://github.com/mluppichini/Shoreline_QGIS_Plugin. More information on the use of this plugin can be found in Supplements.

## 3. Results

### 3.1. DEM and Orthophoto from SfM

The investigated coastline of about 4.2 km was divided into four flights of UAV. We chose to divide the area into four flights for two technical reasons: the first is the autonomy of UAV of about 15 min; the second reason is linked to the size of the DEM and of the orthophotos resulted. Very large grids are difficult to manage and process. The main parameters of UAV surveys are reported in Table 1. The root mean square error (RMSE) between the coordinates of the GCPs and their calculated position in the Metashape models appear in Table 2. The model with the highest error is model 1 with 15 GCPs on a coverage area of 0.15 km². The model with the least number of GCPs is model 4, which is the smallest in terms of coverage area. The number of GCPs of these models is a limiting number that takes into account the study performed by [30]. The DEM resolution is of 2.79 cm/pix for model 1; 2.56 cm/pix for model 2; 2.72 cm/pix for model 3, and 2.67 cm/pix for model 4.

**Table 1.** Main parameters of the four UAV surveys.

| Model | Number of Images | Flying Altitude (m) | Ground Resolution (cm/pix) | Coverage Area (km²) |
|---|---|---|---|---|
| 1 | 361 | 45.3 | 2.79 | 0.15 |
| 2 | 341 | 42.3 | 1.28 | 0.128 |
| 3 | 367 | 38.3 | 1.36 | 0.111 |
| 4 | 561 | 29.3 | 3.48 | 0.060 |

**Table 2.** Root-mean-square error (RMSE) between the coordinates of the GPS control points and their calculated position in the Metashape models.

| Model | Number of GCPs | X RMSE (cm) | Y RMSE (cm) | Z RMSE (cm) | XY RMSE (cm) | Total RMSE (cm) |
|---|---|---|---|---|---|---|
| 1 | 15 | 1.25 | 0.91 | 5.65 | 1.55 | 5.86 |
| 2 | 15 | 0.96 | 0.82 | 4.61 | 1.26 | 4.78 |
| 3 | 9 | 1.73 | 0.60 | 3.35 | 2.83 | 3.82 |
| 4 | 6 | 0.61 | 0.19 | 3.33 | 0.64 | 3.39 |

### 3.2. Shorelines

The development of the algorithm for identification of the shorelines on the basis of topography has led to the use of an $R^2$ threshold above which there is a transition from a profile containing the sea to an exclusive land. During each step, the algorithm makes a shift along with the coast profile of 0.001 m.

Figure 4 shows three frames of the shorelines obtained with DGPS points and UAV image processing. The two shorelines are different, but it is difficult to claim whether one is better than the other. The DGPS-derived shoreline in Figure 4a approximates the real shoreline better than the UAV-derived shoreline. However, in Figure 4c, the behaviour is opposite, and in Figure 4b, the two shorelines approximate the real shoreline better alternating.

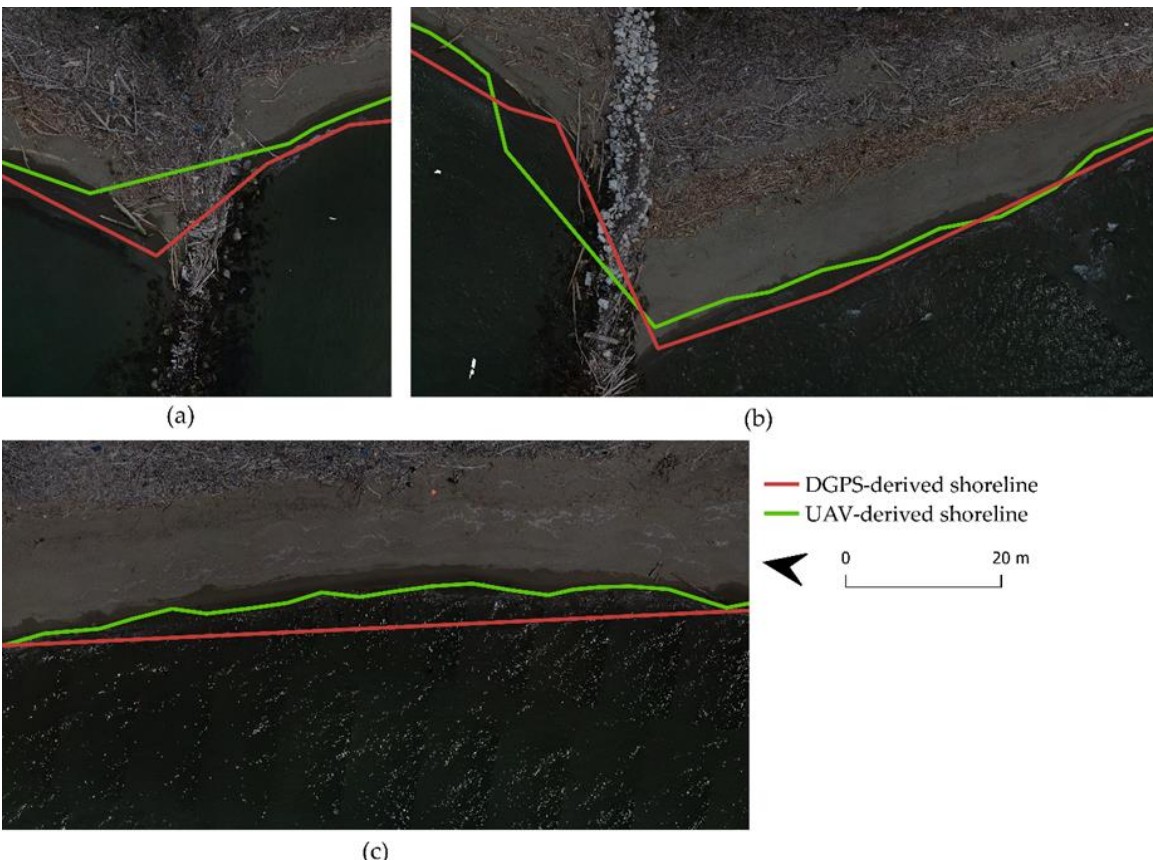

**Figure 4.** Shorelines derived from DGPS points (line red) and from UAV image processing (green line). (**a**) The DGPS-derived shoreline (red line) approximates the real shoreline better than the UAV-derived shoreline (green line); (**b**) the two shorelines approximate the real shoreline better alternating; (**c**) the UAV-derived shoreline (green line) approximates the real shoreline better than the DGPS-derived shoreline (red line).

## 4. Discussion

We evaluated a new method to extract UAV-derived shorelines by comparing the location distance of coastline points along the 112 transects derived from the DGPS points used to realize the coastline that we considered the most flawlessly obtainable. To apply the method, we had to locate transects orthogonal to the beach. Figure 4 shows the distances between the relative points of shoreline derived from DGPS and those derived from UAV images. We needed about 8 transects every 100 m to obtain a minimal error between the two types of shorelines (Figure 5). The minimal mean error with more than 12 transects/100 m is 1.58 m. The number of transects necessary to obtain a precise shoreline is also influenced by the coastline profile; for example, a more irregular coastline needs a greater number of transects. The use of UAV images to extract a shoreline allows you to decide the number and position of transects after the survey. This is not possible when we build the shoreline by using DGPS points: We have to decide the number and the location of point pairs during the sampling phases.

Figure 6 shows the differences in terms of areas by comparing the DGPS-derived with the UAV-derived shoreline. The orange polygons represent the total area when the DGPS-derived shoreline is less seaward than the other shoreline. The blue polygons show the total area, when the DGPS-derived shoreline is more seaward than the other shoreline. The shoreline derived from the UAV images is closer to the beach than the shoreline derived from the DGPS points (Figure 6b). In other words, the UAV-derived shorelines overestimate the mainland compared to the DGPS-derived shorelines. In some cases, equal to about 30% of the total investigated area, the UAV-derived shorelines underestimate the mainland compared to the shoreline derived from DGPS (Figure 6b).

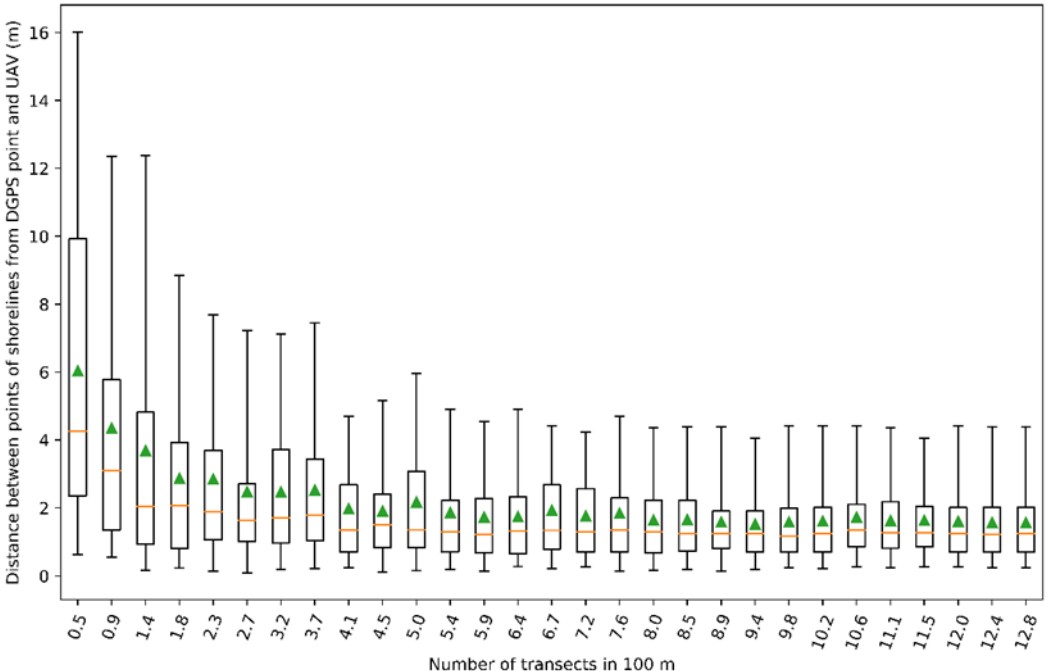

**Figure 5.** Analysis of errors between shoreline points derived from DGPS points and those derived from DEM by Structure from Motion (SfM) processing. The errors are the distance between two relative points of shorelines along 112 transects derived from DGPS points. The box represents the 25th and 95th percentiles, the orange line the median, the green triangle the mean, and the whiskers the 5th and 95th percentiles.

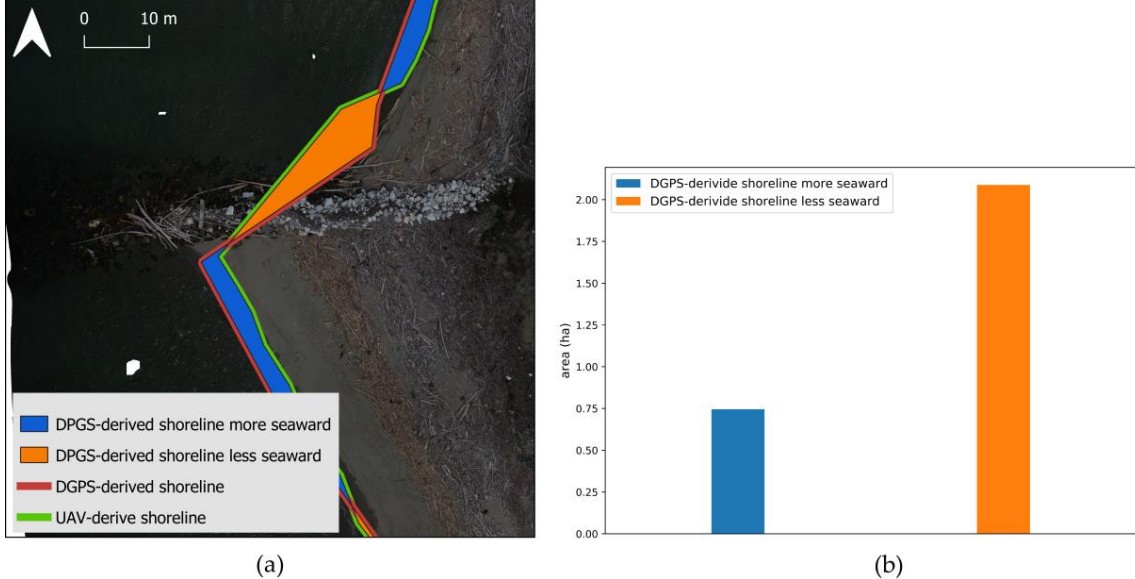

**Figure 6.** Analysis of the differences between DGPS-derived shorelines and UAV-derived shorelines. (**a**) Example of the differences in the areas between DGPS-derived shorelines and UAV-derived shorelines; (**b**) bar plot of the different areas of the beach comparing the two types of shorelines. The blue and the orange rectangles show the total area when the DGPS-derived shoreline is more or less seaward compared to the other shoreline obtained from UAV images.

The RMSE between the DGPS shoreline and the UAV-derived shoreline using 12.8 transects/100 m is 1.69 m, much lower than the methods involving the analysis of satellite images, whose order fluctuates between 6 and 12 m depending on the techniques and images used [5,7,31].

Figure 3c shows the main errors that occur when using DGPS points to build the shoreline. The samplers misinterpolated this stretch of beach by taking an insufficient number of points. The result is a too simplified shoreline, which is a typical error when using DGPS to create a topography profile. Once the data have been sampled, there is unfortunately no possibility of correcting this simplification. On the contrary, in the case of UAV-derived shorelines, we can improve the approximation of the real shoreline by increasing the number of transects. This operation is not linked with the sampling phase of the data, and therefore, it can be applied at any time.

## 5. Conclusions

The method proposed in this study was found to be a valid alternative to the classical methods of shoreline identification based on topography. This method makes it possible to obtain shorelines using the topography obtained from UAV images; it is a novelty compared to other uses of DEMs obtained from UAV images present in the literature. Several studies have used high resolution DEMs to obtain differences in volume between sequential samples over time. These methods do not allow to compare the results with those obtained with other techniques such as shorelines obtained from satellite and/or aerial images. Our method made it possible to overcome this limit with the identification of a margin between water and land.

This approach is innovative, and it could also be a valid alternative to the methods based on manual identification or on remote-sensing image colours. In this respect, it is very difficult to compare differently-derived shorelines when the errors are about 1–2 m. When we compare the use of Satellite images and of DGPS, identification of the error between the two methods is simpler than when we compare the DGPS-derived shoreline with UAV-derived images. This happens because the error of DGPS points to extract the shoreline is negligible compared to the errors that occur when using satellite images with a pixel size of about 10 m. However, when we compare DGPS-derived shorelines with UAV-derived images, all the errors are of the same order of magnitude. This work has shown that in some cases the DGPS-derived shoreline is better than the UAV-derived shoreline, but in the same number of cases, the roles are reversed. Therefore, it is very difficult to determine the best method by a simple comparison of the errors. However, this new method has two main advantages regarding the use of DGPS points. The first advantage is the amount of time it takes to obtain a stretch of coast: a UAV takes less than a DGPS. To sample 4.5 km of coastline with UAV, we took about 3 h while with the DPGS, we took about 6 h. The second advantage is that the position of the transects used to reconstruct the shoreline can be decided after sampling and not during acquisition of the DGPS points.

**Supplementary Materials:** The following are available online at http://www.mdpi.com/2073-4441/12/11/3110/s1, Figure S1: Installing Plugin from ZIP utility of QGIS software. Figure S2: Shoreline Identification Plugin interface.

**Author Contributions:** Conceptualization, M.L. and M.B.; methodology, M.L., M.B., M.P. and A.B.; software, M.L.; validation, M.L., M.B. and M.P.; formal analysis, M.L. and M.B.; investigation, M.L. and M.B.; resources, M.B., M.P. and A.B.; data curation, M.P., M.L. and A.B.; writing—original draft preparation, M.L. and M.B.; writing—review and editing, M.P., A.B. and S.M.; supervision, M.B.; project administration, M.B., M.P., A.B. and S.M. All authors have read and agreed to the published version of the manuscript.

**Funding:** This research received no external funding.

**Acknowledgments:** We thank the "Migliarino-San Rossore-Massaciuccoli Natural Parks" for supporting our work. Part of this study has been developed within the framework of the project "Cambiamenti globali e impatti locali: conoscenza e consapevolezza per uno sviluppo sostenibile della pianura apuo-versiliese" awarded to M.B. and funded by the Fondazione Cassa di Risparmio di Lucca, call 2019 and "PRA-2018-41 Georisorse e Ambiente" funded by the University of Pisa.

**Conflicts of Interest:** The authors declare no conflict of interest. The funders had no role in the design of the study; in the collection, analyses, or interpretation of data; in the writing of the manuscript; or in the decision to publish the results.

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
