# Peer review of "A New Beach Topography-Based Method for Shoreline Identification"

_water, doi:10.3390/w12113110_

Round 1

Reviewer 1 Report

Thank you very much for your paper. I enjoyed reading it. Here are my remarks:

18: What is the difference beteen cameras and drones? You used a camera on your drone.

23: Does SfM work only on sandy beaches?

66: Does have LiDAR really a lower resolution?

75/76: Two times "obtained"

Chapter 2.1 is too short. No information about the length of the stretch, etc

90 ff: Maybe passive time is better? Your choice.

90: Why did you choose 224 points and not more? In Fig.4 you could improve the DGPS reults by using more points significantly.

119: ???

135: How many GCPs did you use? What is a GCP and how did you determine the coordinates? Accuracy of the coordinates?

145: soil?

155/156/158: Do you mean transects?

156/161/163/209/217/270:remove blanks

166: Where are the Supplements?

168: "T"he location ..

181/190: "m"etashape

183: How many GCPs did you use for each model?

184 and Table 1: 2.79 does not fit with 1.39 aso. What is the differece between these two resolutions?

194: R2?

194: Name/quantify thresholds

195: The algorithm makes .... I do not understand this sentence.

198: Why is that so bad? You have a ground resolution in cm and e.g. in Fig 4(a) the green line is is not fitting the shoreline at all? The distance to the water is more than 10m.

207: ..most perfactly..??

208: Do you mean Figure 4?

218/219: Do you mean "areas"? I do not see rectangles in Fig.6(a)

227: .. images, whose ..

260: 1-2m? Where can I reproduce that in your text? How did you consider the different times of data aquisition? Is there any influence of tides?

269: Can you quantify the time? How long did it take to perform the two different approaches?

The paper is quite short. It might be good to describe the methodology more detailed and also the results and discussion.

Author Response

We thank the reviewer for his contribution to improve the paper.

Reviewer 2 Report

In the abstract I think line 30 to 36 are not necessary and could be deleted

Line 118. What is the meaning of this sentence: “All acquisitions were obtained with a fixed zoom system???,

Line 124. Is the abbreviation AGL is introduced before?

In the PDF file line 195 to 200 and 229 error in reference link

I think the new method introduced in this paper is interesting and innovative. However, application of this method might be complicated in other coastal areas where tidal range is much higher than Mediterranean.

Author Response

We thank the reviewer for his contribution to improve the paper
